# Impact of Depsychopathologization of Transgender and Gender Diverse Individuals in ICD-11 on Care Delivery: Looking at Trans Expertise through a Trans Lens

**DOI:** 10.3390/ijerph192013257

**Published:** 2022-10-14

**Authors:** Anna Baleige, Mathilde Guernut, Frédéric Denis

**Affiliations:** 1EA 75-05 Éducation Éthique Santé, Faculté de Médecine, Université François-Rabelais Tours, 2 Boulevard Tonnellé, 37044 Tours, France; 2UMR 8163 Savoirs, Textes, Langage, Centre National de la Recherche Scientifique, Université de Lille, Rue du Barreau, BP 60149, 59653 Villeneuve d’Ascq, France; 3Service D’odontologie, Centre Hospitalier Universitaire de Tours, 2 Boulevard Tonnellé, 37044 Tours, France

**Keywords:** transgender persons, health services needs and demand, knowledge, community-based participatory research, international classification of diseases

## Abstract

Depsychopathologization of transgender and gender diverse (TGD) individuals in the eleventh revision of the International Classification of Diseases (ICD-11) called for a shift in care delivery models, based on free and informed consent. Public health policies face epistemic and discriminatory challenges and consensus built on evidence-based data is needed. TGD communities were consulted but did not actively participate in ICD-11 and the following public health debates. There is a need for TGD perspective—both in research and practice. This study draws on a peer-led participatory approach and explores TGD participants’ recommendations based on unexploited French data from ICD-11, in which 72 TGD gave feedback on public policies. Lexicometric analyses were conducted using the ALCESTE method and resulted in a two-step double Descending Hierarchical Classification. Sex, gender, and health consumption were analyzed as secondary variables. The first classification highlighted five main topics: care pathways, training of professionals, access, literacy, and civil status change, developed into 12 targets in the second classification. While sex and gender appeared to have little impact on discourses, recommendations varied according to received care. This study supports the growing scientific consensus of a public health approach to face TGD health challenges and emphasizes TGD individuals’ expertise.

## 1. Introduction

Transgender and gender diverse (TGD) [1] individuals face systemic barriers in accessing health services [2]. Simultaneously, studies highlight the significant health needs of this population [3]. To address them, it is proposed to transform health services organization, based on the free and informed consent model, and to develop trans-affirmative and non-discriminatory health services [4].

The shift in care delivery models is partly based on the depsychopathologization of TGD persons in the eleventh revision of the International Classification of Diseases (ICD-11) [5]. ICD-11 brought along epistemic transformations characterized by a politicization of classificatory considerations [6,7] while maintaining an appearance of neutrality focused on classification comparisons [6,8,9]. Debate between diagnoses of Gender Dysphoria and Gender Incongruence focused on whether suffering at the Index Period was necessary as a diagnostic criterion [9]. Index Period was defined as “a period of time […] during which gender incongruence, distress and dysfunction may have been particularly prominent” [10]. ICD-11 outperformed, both practically [9] and conceptually, as field studies conducted with the World Health Organization (WHO) highlighted that psychological distress was not systematic and primarily arose from widespread transphobia [9,10,11], interpreted as minority stress [10,12]. The non-necessity and external origin of psychological distress supported the WHO process of depsychopathologization [10]. In this context, depsychopathologization is best understood as part of a paradigm shift, rather than a natural evolution of medical epistemologies [8]. Thus, it appeared both as a violation of basic human rights [7] and as an unnecessary burden on the lives of TGD individuals and clinical practice [4,5,9].

Out of the six field studies conducted for ICD-11 [9], one was carried out in France in a multidisciplinary primary healthcare provider (Maison Dispersée de Santé, now MDS), based on risk-reduction and the free and informed consent model [4], and working closely with local TGD organizations [5,10]. This study drew on a participatory aspect. It was indeed noted that TGD communities, although consulted, were voluntarily kept away from decisions [6]. This study was carried out in partnership with a WHO Collaborative Center in mental health—their terms of reference include the full participation of service users in research and policymaking [13]. Moreover, the study included several TGD consultants and was extended with a questionnaire developed with the local communities [10,14].

However, while these dynamics demonstrated the possibilities of providing free and informed consent-based health care [5,10], the study did not address power dynamics inherent to participatory research [15], especially on such a politicized subject [16]. In fact, beyond the systemic transphobia built into the research pipelines [14], incongruence between TGD consultants and research professionals was documented during the study’s implementation and subsequent stages [14], leading to their gradual, then definitive, withdrawal [9,10,14].

These issues may appear abstract and remote from everyday experience—especially since ICD-11 has not resulted in a massive transformation of health systems, but rather in political support for local and activist initiatives [6]. This is all the more concerning as most health systems include built-in transphobia [2], especially in France [17].

Given health needs [3] and challenges of transforming practices and systems [18], developing and consolidating culturally appropriate knowledge within a human rights framework seems all the more necessary. Under-representation of TGD individuals in research contributes to the lack of quality data [19], both linked to TGD researchers’ better awareness of transphobic biases in research and their deeper understanding of TGD lives and experiences. Similarly, the WHO argued for over a decade that care safety is improved by service users’ participation [20].

We hypothesize that a study led by a TGD researcher will provide results with a better ecological validity and add to the existing literature on organizational impacts of TGD health needs. This research is in line with survivor research and addresses a number of transversal epistemic challenges [16]. As such, this paper follows this dynamic in order to contribute to public health policies by providing insights from directly impacted people. Its main objective is to report on recommendations from participants of the study conducted in France as part of ICD-11 and contribute to build a shared knowledgebase between professionals and TGD communities. Secondarily, this paper also addresses the impact of sex assigned at birth (SAAB) and gender, as well as distress and health consumption, on the recommendations. It draws on unexploited data to contribute to reclaiming knowledge that communities originated.

## 2. Materials and Methods

### 2.1. Epistemic Challenges

This paper aims to highlight ways of improving health care systems for TGD individuals based on their experiential knowledge, i.e., their subjective experience [21]. Knowledge is socially constructed through language by means of production and interpretation, ideological frameworks, and paradigms [16,22]. Analyzing knowledge requires familiarity with the socio-cultural context and therefore involves the subjectivity of analysts—a known factor in research power dynamics [15,23]. To limit these biases, widely present in research on TGD health [8], and improve the quality of the analyses [15,16], this paper draws on a participatory approach, following survivor research, i.e., under the direction of someone directly concerned [24].

As translation is an inherently important part of the research process, and following current recommendations [25], all translations were conducted simultaneously by two translators (B.A. & G.M.), and discrepancies were answered by consensus building. Both translator-researchers are native speakers of French (France, continental), have excellent English skills, and are familiar with concepts specific to health and TGD individuals [26]. This approach differs from back translation commonly used in medical research and aims at improving ecological validity and reducing biases [25].

### 2.2. Ethical Conformity

The protocol complied with the Helsinki Declaration of 1975—as revised in 2008—and the WHO Good Clinical Research Practice guidelines. The research received approval from biomedical research French authorities: the Comité de Protection des Personnes Nord-Ouest IV, the Comité Consultatif sur le traitement de l’information en matière de Recherche dans le domaine de la Santé, and the Commission Nationale de l’Informatique et des Libertés. Informed written consent was obtained from every participant.

Data access was provided through a free of charge convention with the center.

### 2.3. Materials

Data were extracted from the 2017 study conducted at MDS in collaboration with WHO. Participation to the study was proposed during routine professional appointments and, following a presentation of the study, those interested were directed to the inclusion step [10]. Seventy-two transgender individuals receiving care at MDS voluntarily participated in the study. They were interviewed using two questionnaires: a main questionnaire translated and adapted from the one used in Mexico [11], and an additional one specific to the French study [10]. The first questionnaire was specifically built for the studies on ICD-11, while the additional one was co-constructed with local TGD communities and specific to France. Answers to the first questionnaire were used to gather quantitative socio-demographic and health consumption information on participants.

Recommendations from participants were collected from the additional questionnaire. Participants were asked two consecutive questions: “do you have any proposal to improve the information, support, or care provided to persons in transition?”; “do you have any proposal for health professionals to better respond to the needs of persons in transition?”. Responses were open-ended and non-mandatory and formed two sets of recommendations.

Sex and gender data were collected via a two-step method [27], participants could choose their SAAB (female, male, intersex) and gender from a list (woman, man, transgender woman, transgender man, genderqueer, intersex) or answer openly. Group analyses in the original publication excluded three participants who openly answered and based the analyses on SAAB [10]. We included all participants and categorized them into three groups: male, female, and gender diverse [1]. Considering current evidence of TGD health behaviors [28], we conducted two sets of lexical analyses, one classifying transgender women and men in gender diversity, and one in a binary manner. Since this paper also questions categorization of gender diverse individuals based on their experience of healthcare, we expanded the initial descriptive analyses [10] and included data on access to health services.

For health access data, we altered the initial categories in accordance with current evidence on TGD healthcare [3]. Three categories emerged, organized around time and motive: medical transition, healthcare use following psychological distress at Index Period, and life-long mental healthcare use. Regarding life-long categories, participants were asked whether they had accessed psychological support, hormone treatment, surgery, or other health services. For psychological distress at Index Period, we analyzed three successive yes/no questions: “have you experienced psychological distress related to your gender identity?”; If yes: “have you sought specialized mental health care to deal with this?”; If yes: “did you receive this treatment?”. For each, we calculated the delay between study participation and last service access/Index Period.

### 2.4. Analyses

All statistical analyses were performed using R v4.2.0. Descriptive statistics included frequencies (in percentages) for categorical variables, means, standard deviations (SD), and range for continuous variables. Comparisons were based on chi-square tests. For open-ended data, a corpus was created and pre-processed using Notepad++ v8.3.3. Lexical analyses were performed with the IRAMUTEQ software package v0.7alpha2 [29] for R v3.1.2.

Both sets of recommendations were pulled together in a single corpus, each recommendation associated with a set of descriptive variables informing on the set of recommendation (binary), SAAB (binary), gender (two sets of three categories), medical transition (three variables, binary), and psychological care at Index Period (three variables, one binary and two of three categories) or life-long (binary).

The corpus was then pre-processed using the ALCESTE method in three consecutive steps [30]:-Each word’s form in the corpus is recognized and, after an initial part-of-speech tagging, lemmatized, replacing them with the associated lemma.-The corpus is segmented in elementary context units (ecu) following a pre-established set of rules [30], allowing for lemmas to be analyzed in semantic context.-The ecu are segmented to build a grid for analysis.

During Descending Hierarchical Classification, lemmas are ordered based on iterative classifications comparing segments of ecu. Each step splits the corpus in two classes to maximize the chi-square test. The process is repeated until the desired number of classes is reached or the corpus portion included in the analysis is too low. To ensure class stability, two parallel classifications are built based on different size of lexical segments, and then merged. Lemmas are classified in clusters of lexical contexts, which can be analyzed based on word list, or extraction of text segments most representative of the cluster. It also allows to assess specificities of descriptive variables identified in the analysis [29,30].

## 3. Results

### 3.1. Description of Participants

More than half of the participants were assigned male at birth (61.1%, n = 44) and no participant reported an intersex SAAB. Most participants reported a binary gender identity (77.8%, n = 56). 34 (47.2%) identified as “woman”, 22 (30.6%) as “man”, 4 (5.6%) as “transgender woman”, 4 (5.6%) as “transgender man”, and 8 (11%) as “other”. No participant identified as genderqueer or intersex gender. Other genders included “neutral”, “non-binary with male tendency/agender performing male”, “¾ woman, ¼ man”, “gender fluid/non-binary”, “fluid with female tendency”, “80% female androgynous person”, “non-binary but more feminine”, and “non-binary”.

As shown in Table 1, a very large proportion (90.3%, n = 65) of participants have used the healthcare system in relation to their gender identity. Current gender identity and SAAB did not impact the use of the healthcare system (*p* > 0.05), except for other types of care in the case of participants who identified as “women” (χ^2^(2) = 9.5, *p* < 0.05) or with a male SAAB (χ^2^(1) = 10.8, *p* < 0.01). The time frame between the study and the last access to healthcare linked to gender identity was short with an average of 1.5 (2.9; 0–15) years. In contrast, access to healthcare at Index Period was low with only 34.4% (n = 21) of participants who experienced distress getting access to mental healthcare, for them, the delay was 14.0 (8.1; 5–45) years.

### 3.2. Lexical Analysis of Participants’ Recommendations

Out of the 72 participants, 68 (94.4%) provided an average of 3 (1.9; 1–11) recommendations for the first set regarding improving information given, support, or care provided to persons in transition, and 65 (90.3%) provided an average of 2.5 (1.6; 1–8) recommendations for the second set (how health professionals can better respond to their needs).

Both sets of recommendations (n = 133) were pulled together in a single corpus. After lemmatization, the corpus consisted of 5176 occurrences of 1080 unique forms, i.e., a type-token ratio of 20.9%. The hapax, or single occurrence forms (n = 658), made up 60.9% of all unique forms and 12.7% of total occurrences. The algorithm isolated 183 ecu out of the 133 recommendations, which were segmented in units of 12 and 14 lemmas for analysis. Given the corpus’ linguistic homogeneity, we decided to carry out two parallel analyses at different levels of classification.

### 3.3. First Descending Hierarchical Classification

The first Descending Hierarchical Classification focused on size and individuated five stable clusters for both sets of recommendations. Each cluster contains lemmas most associated to a specific lexical context, their distribution and frequency are presented in Figure 1. Of the corpus, 92.9% was analyzed. Clusters 1, 2, and 3 focus on healthcare and make up for 57.0% of the analyzed content. The most prominent clusters are clusters 4 (33.5%), 1 (27.6%), and 3 (20.0%) respectively.

Cluster 1 is associated with lemmas related to TGD individuals’ health pathways: “take” and “care”, “surgery”, and “depilation”. It also focuses on ways to improve them: “difficult”, “problem”, and “real”. All these associations, as well as the one presented in Figure 1, are statistically significant (*p* < 0.001). As such, recommendations associated with this cluster aim at improving health pathways for transgender persons. Cluster 2 (9.4%) is associated with the “training” of “health” “professional” (*p* < 0.001). It also focuses on health pathways and in particular “specialize” (*p* < 0.01) and “depsychopathologize” (*p* < 0.05) services. Contrary to cluster 1 where recommendations mostly focus on new ways to improve pathways, cluster 3 focuses on lifting existing barriers. It is associated with lemmas representing problems such as “long”, “mandatory” (*p* < 0.001), “time” (*p* < 0.01) and “waiting” (*p* < 0.05). Cluster 4 is outside the healthcare system and its recommendations aim to raise awareness in civil society about TGD people’s rights and issues. Associated lemmas include specific intervention targets such as “school”, “media”, “middle school” (*p* < 0.001), and “high school” (*p* < 0.01); but also means to raise awareness, including “talk”, “intervention”, “raise awareness”, “inform” (*p* < 0.001), or “develop” (*p* < 0.01). Cluster 5 makes up for the last 9.4% of the analyzed content and focuses on administrative procedures—it is associated with lemmas representing the already known notions of “civil” “status” “change” (*p* < 0.001); the “law” and procedures regarding “certificate” for which you have to “pass” in front of a “psychiatrist” (*p* < 0.001). Examples of recommendations including associated lemmas in context for each cluster are presented in Table 2.

SAAB and gender were not associated with any specific cluster, regardless of gender classifications. Cluster 4 “raise awareness in civil society” was associated with the first set—improving information, support, or care provided to persons in transition (*p* < 0.001)—while the second set—how health professionals can better respond to their needs—was linked to clusters 2 “improve training of health professionals”, 3 “reduce barriers in access to care” (*p* < 0.01) and 1 “improve care pathways for transgender persons” (*p* < 0.05). Cluster 5 “facilitate the change of civil status” was not associated to any set (*p* > 0.05).

Life-long service uses were not associated with any specific cluster. Nonetheless, patterns seem to emerge, opposing life-long psychological support and medical transition (Figure 2). At Index Period, experience of psychological distress was associated with cluster 3 (χ^2^(1) = 6.3, *p* < 0.05) and not having to seek mental health care was associated with cluster 1 (χ^2^(1) = 12.0, *p* < 0.001).

### 3.4. Second Descending Hierarchical Classification

The second Descending Hierarchical Classification focused on precision and resulted in twelve stable clusters for both sets of recommendations using 76.5% of the corpus. Clusters were put together according to their ascendant in the first classification, resulting in five groups (Figure 3).

The first group emerged from Cluster 1 and consisted of five clusters focusing on depsychopathologizing (Cluster 1.0; 5.0%), improving communication (Cluster 1.1; 7.9%), increasing access to affordable surgeries (Cluster 1.2; 5.7%), training medical staff (Cluster 1.3; 5.7%), and facilitating paperwork (Cluster 1.4; 5.0%). Another group of two clusters emerged from Cluster 3: reducing waiting and assessment periods (Cluster 3.0; 12.9%), and changing official teams (Cluster 3.1; 10.7%). Cluster 4 was split, forming a final group of three clusters: education in medias (Cluster 4.0; 6.4%), schools (Cluster 4.1; 13.6%), and improving global education and prevention (Cluster 4.2; 4.3%). Unlike other clusters, Clusters 2 and 5 were not divided during the second classification, resulting in two similar clusters (Cluster 2.0 and 5.0), both consisting of 11.4% of the analyzed corpus.

Female SAAB was associated with Cluster 4.0 (promoting education in medias—χ^2^(1) = 9.2, *p* < 0.01), and gender diversity with Cluster 1.4 (facilitating paperwork), both when including or excluding transgender men and women (χ^2^(1) = 4.9, *p* < 0.05 vs. χ^2^(1) = 12.9, *p* < 0.001). The first set on improving information, support, or care provided to persons in transition remained associated with recommendations from Clusters 4.0 (χ^2^(1) = 4.8, *p* < 0.05) and 4.1 (χ^2^(1) = 19.1, *p* < 0.001), and the second set with Clusters 1.2 (χ^2^(1) = 5.8, *p* < 0.05), 2.0 (χ^2^(1) = 8.8, *p* < 0.01), and 3.0 (χ^2^(1) = 8.2, *p* < 0.01).

Life-long service uses for medical transition regarding surgery and other care was associated with Cluster 1.0 (depsychopathologization, χ^2^(1) = 3.9, *p* < 0.05; χ^2^(1) = 4.5, *p* < 0.05). Psychological distress at Index Period was associated with cluster 3.0 (reducing waiting and assessment periods, χ^2^(1) = 4.9, *p* < 0.05) and not having to seek mental health care was associated with cluster 1.3 (training medical staff, χ^2^(1) = 10.2, *p* < 0.01).

The second classification clarified the components of initial clusters at the expense of losing part of the corpus (Table 3). Clusters 2 and 5 did not lose any data between the two classifications, while Clusters 1, 3, and 4 lost respectively 3.5%, 0.6%, and 13.5% of the initially analyzed data, showing recommendations existing inside the initial clusters, but outside of secondary ones.

## 4. Discussion

This study highlights the deep and insightful understanding of the health systems among TGD service users, needed to face current challenges. The population sample for a qualitative study supports the validity of the results, especially since they appear valid for a large number of Western health systems, and despite containing elements specific to the French European context.

The main objective of this paper leads to the observation that the process of depsychopathologization is not part of a natural evolution of practices, but a shift in how the problem is addressed [8,31]. The organizational impact of the depsychopathologization process [5], initiated by the WHO, appears to be part of a paradigm shift [32]. For TGD individuals, several dominant paradigms in science are questioned, including sexual difference [8,31]. From the care pathways of a small number of mentally ill individuals, the debate has shifted to the global health promotion of a vulnerable and growing population—definitively establishing the health of TGD individuals as public health-related [3,4].

In the aforementioned French study [10], the research team did not take into account the power issues inherent in participatory research [15], leading to methodological discrepancies [33]. Paradigm shifts impact ideological frameworks in research [22], leading to methodological and political challenges [33]. This places community-led participatory research in a key position to tackle this paradigm shift [16]. Our methodology based on lexicometry [34,35] and participatory design [15,16] also contributes to ecological validity. One of the main limitations of lexical analyses is their difficulty accounting for discourse content when faced with a diverse vocabulary encompassing similar semantic notions [34]. Here, the relative lexical homogeneity of the corpus allowed us to partially overcome this limitation. The results recontextualization stage also involves the analysts’ subjectivity [15,23], also partially controlled by the participatory design [15,16].

While lexical homogeneity allowed for good quality of analysis, it remains unexplained. However, it underlies a conceptual homogeneity allowing to classify almost all recommendations into five main categories: improve care pathways, train professionals, reduce barriers, facilitate civil status change, and raise awareness in civil society. The second analysis revealed the conceptual stability of these categories, apart from Cluster 4—representing the greatest loss of content in the entire analysis (Table 3). Thus, while the media, schools, and general prevention strategies emerged as prime targets, a plurality of opinions seems to persist on remaining interventions.

Participants’ recommendations align with recent public health literature [3]. Improving health pathways is based on depsychopathologization [4,5,7] and calls for improving communication, access to affordable surgeries, training of staff, and administrative procedures. Professional training is also a sufficiently salient element to constitute an autonomous cluster [36]. Barriers in access to care seem equally based on the perceived excessive waiting time and the inadequacy of current dedicated services. Finally, civil status change appears as another administrative challenge. In this regard, France did not have any dedicated procedure for civil status change before 2016—following Turkey’s conviction by the European Court of Human Rights for similar events [37]. The law came into effect in 2017 and, since then, civil status change became free of charge and does not require a medical assessment but remains at the discretion of the judge—which is widely criticized by TGD organizations.

SAAB and gender do not appear as organizing elements in the participants’ discourse. Difficulties related to changing one’s civil status are associated with gender diverse individuals, which is consistent with France’s administrative confusion of sex and gender and its lack of a third administrative gender as an alternative to binarity [28]. The association between individuals with female SAAB and promoting education in the media, although strong, does not appear obvious and would deserve a specific exploration.

Although both sets of recommendations do not split the discourse, it appears polarized between transforming the health care system on one hand, and society as a whole on the other. This convergence of discourses is part of a global vision of health and is consistent with public health strategies [3]. The emergence of a shared community discourse centered on human rights [6,7] could be an explanatory element for both conceptual homogeneity and strong compatibility with health promotion approaches.

Personal experience of the healthcare system also had minimal impact on discourse construction. Depsychopathologization appears more closely associated with participants using surgery or other specific care for medical transition (hair removal, etc.). This may be interpreted in context by the scarcity of trans-affirmative professionals and the widespread requirement for psychiatric evaluations to access care. Psychological distress at Index Period is associated with reducing waiting and evaluation periods—a recurring criticism of the French mental health system shared by cisgender individuals. Finally, the strongest association is between participants who did not experience psychological distress at Index Period and training health professionals—referring directly to the epistemic issue raised by the WHO: cisgender professionals’ belief of a necessary suffering [8,9,10,11].

## 5. Conclusions

This study supports the growing scientific consensus of a public health approach to face TGD health challenges [3]. TGD individuals and perspectives are underrepresented in research [8,19]. As critics arise on the lack of WHO support towards system change, leaving participatory data unexploited is all the more questionable and calls for a better understanding of political dynamics in medical epistemology and systemic transphobic biases in research [15,16,23]. Given epistemic uncertainties, critical studies and critical discourse analysis should continue to be promising areas of research [38].

## Figures and Tables

**Figure 1 ijerph-19-13257-f001:**
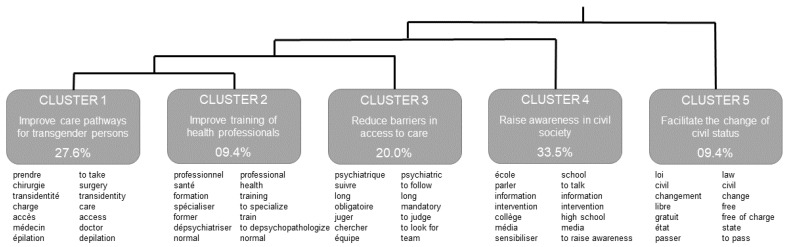
Descending Hierarchical Classification of 92.9% of the corpus. Correlations are based on chi-squared test cluster vs. remaining corpus. Significance level for lemmas is *p* < 0.05 with up to seven lemmas and their English translation illustrating each cluster.

**Figure 2 ijerph-19-13257-f002:**
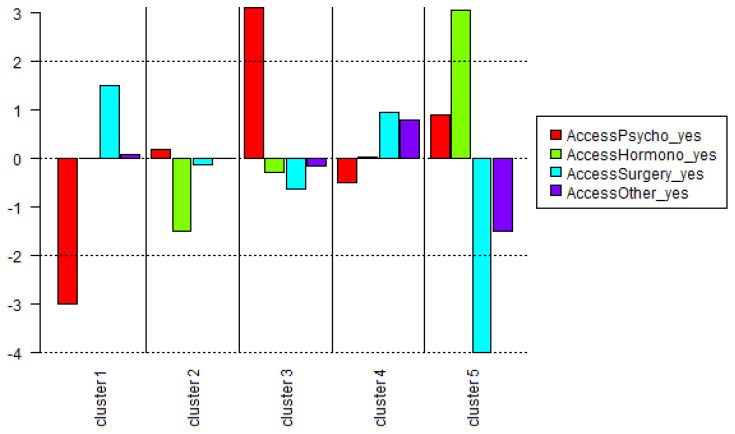
Specificities of descriptive variables describing life-long care between clusters identified by the first Descending Hierarchical Classification. Chi-square presented on ordinate axis; over-representation is represented as positive, under-representation as negative. No correlation is statistically significant. AccessPsycho/AccessHormono/AccessSurgery/AccessOther—Life-long care for psychological support/hormonal treatment/surgery/other (yes; no).

**Figure 3 ijerph-19-13257-f003:**
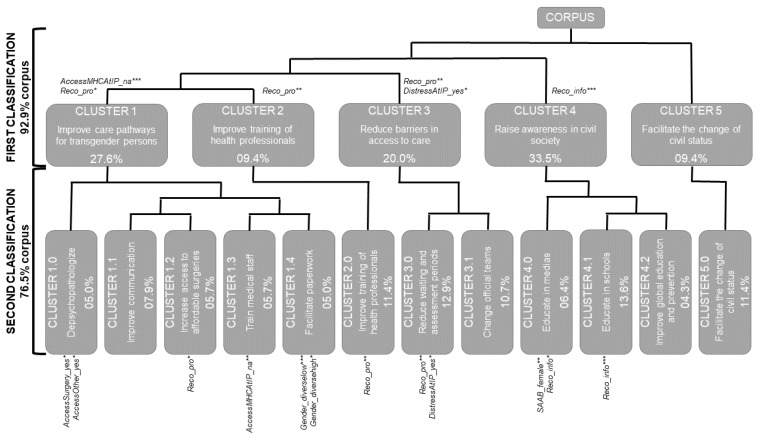
Two-step Descending Hierarchical Classifications.Modalities of variables are statistically associated with the cluster (* *p* < 0.05; ** *p* < 0.01; *** *p* < 0.001). variable—description of variable (modalities = description of modality): AccessMHCAtIP—Use of psychological support at Index Period (yes; no; NA); AccessSurgery—Life-long care of surgery (yes; no); AccessOther—Life-long other types of care related to medical transition (yes; no); DistressAtIP—Psychological distress at Index Period (yes; no); Gender—Gender (womanhigh; manhigh; diversehigh = gender categories with transgender men and women grouped as diverse; womanlow; manlow; diverselow = gender categories with transgender men and women considered as binary); Reco—Sets of recommendations (info; pro); SAAB—Sex Assigned At Birth (female; male; intersex).

**Table 1 ijerph-19-13257-t001:** Age, education, life-long care, and psychological distress at index period linked to gender identity according to current gender identity (women vs. men vs. gender diverse).

	Total Sample (n = 72)	Women (n = 34)	Men (n = 22)	Gender Diverse (n = 16)
	Mean (SD; Range)			
Age (years)	27.7 (9.7; 18–50)	31.6 (11.6; 18–50)	22.6 (4.5; 18–34)	26.2 (6.8; 18–47)
Education (years)	13.7 (2.5; 9–20)	13.6 (2.6; 9–20)	13.4 (2.5; 9–18)	14.7 (2.3; 10–17)
**Life-long care linked to gender identity**				
	n (%); Mean (SD; Range)			
Any healthcare	65 (90.3%)	31 (91.2%)	20 (90.9%)	14 (87.5%)
Delay before study (years) ^a^	1.5 (2.9; 0–15)	1.8 (3.6; 0–15)	1.3 (2.5; 0–10)	1.1 (1.0; 0–3)
Psychological Support	34 (47.2%)	18 (52.9%)	9 (40.9%)	7 (43.8%)
Delay before study (years) ^a^	3.7 (4.1; 0–19)	4.6 (5.1; 0–19)	2.7 (2.6; 0–6)	2.7 (2.4; 0–6)
Hormonal treatment	60 (83.3%)	28 (82.4%)	19 (86.4%)	13 (81.3%)
Delay before study (years) ^a^	2.8 (5.4; 0–27)	4.1 (7.3; 0–27)	1.8 (3.0; 0–11)	1.5 (1.3; 0–4)
Surgery	23 (31.9%)	11 (32.4%)	8 (36.4%)	4 (25%)
Delay before study (years) ^a^	3 (4.7; 0–17)	4.0 (6.1; 0–17)	2.5 (3.6; 0–10)	1.3 (1.0; 0–2)
Other	25 (34.7%)	20 (58.8%)	1 (4.5%)	4 (25%)
Delay before study (years) ^a^	2.4 (4.4; 0–16)	2.9 (4.9; 0–16)	0.0	0.8 (1.0; 0–2)
**Psychological distress linked to gender identity at Index Period**				
	n (%); Mean (SD; Range)			
Age at Index Period (years)	10.7 (7.1; 3–40)	11.8 (7.7; 3–40)	9.3 (4.8; 3–22)	10.4 (8.5; 4–40)
Experience of psychological distress	61 (84.7%)	27 (79.4%)	21 (95.5%)	13 (81.3%)
Sought mental health care ^b^	21 (34.4%)	7 (25.9%)	10 (47.6%)	4 (30.8%)
Received mental health care ^c^	21 (100.0%)	7 (100.0%)	10 (100.0%)	4 (100.0%)
Delay before study (years)	14.0 (8.1; 5–45)	16.4 (13.0; 5–45)	11.5 (3.6; 7–17)	15.8 (4.6; 10–20)

^a^. For participants with multiple accesses, the delay is based on the last access to healthcare; ^b^. % out of participants with an experience of psychological distress; ^c^. % out of participants who sought mental health care.

**Table 2 ijerph-19-13257-t002:** Recommendations from participants to improve the information, support, or care provided, and response from health professionals.

Cluster	Representative Quotes (Gender; Set)
**1. Improve care pathways for transgender persons**	*Changer la classification dans la **CIM **** pour ne plus être obligé de passer par les psychiatres.* Change the classification in the **ICD **** so that you don’t have to go through the psychiatrists anymore. (man; how health professionals can better respond)
	*Développer les **soins **** et structures pour les trans (**chirurgie ***, épilation ****, etc…).* Develop **care ***** and structures for trans people (**surgery ***, hair removal **,** etc...). (woman; improving information, support, or care provided)
	*Ne pas forcer les personnes à la **prise ***** d’hormones.* Do not force people to **take ***** hormones. (man; how health professionals can better respond)
	***Faciliter *****l’**accès ***** à l’ALD avec l’instauration d’un **vrai **** protocole.* **Facilitating * access ***** to long-term healthcare coverage with the establishment of a **real **** protocol. (man; how health professionals can better respond)
**2. Improve training of health professionals**	*Plus de **formation ***** pour les **professionnels *****, peut-être plus de moyens, mais le manque de **formation ***** est le principal.* More **training ***** for **professionals *****, maybe more resources, but the lack of **training ***** is the main one. (gender diverse; how health professionals can better respond)
	*Avoir une **formation ***** des **professionnels ***** sur le comportement à avoir quand on reçoit une personne trans.* **Train *** professionals ***** on how to behave when dealing with a trans person. (man; how health professionals can better respond)
	***Former *****les **professionnels ***** de **santé ***** à la question pour pouvoir repérer les personnes plus tôt pendant l’enfance ou l’adolescence.* **Train * health care professionals ***** on the issue so that they can identify individuals earlier in childhood or adolescence. (man; how health professionals can better respond)
	***Former *****les **professionnels ***** de **santé ***** uniquement par des personnes trans.* **Train * health care professionals ***** only by trans people. (gender diverse; how health professionals can better respond)
**3. Reduce barriers in access to care**	*Éviter de trop creuser en consultation (ex: **chercher ***** une raison dans l’enfance) car quand on va **voir *** un **psychiatre *** le questionnement est là depuis longtemps donc il **faut ***** éviter de nous remettre en question car ce n’est pas une maladie.* Avoid digging too deep in consultation (e.g., **looking for ***** a reason in childhood) because when we go to **see*** a **psychiatrist *** the questioning has been there for a long time so it is **necessary ***** to avoid questioning ourselves as it is not an illness. (man; how health professionals can better respond)
	*Informer les personnes trans’ sur le fait que l’**équipe *** officielle ***** n’est pas un passage **obligatoire ***.*** Inform trans people that the **official *** team ***** is not a **requirement *****. (woman; improving information, support, or care provided)
	*Deux **ans **** de psychiatrie c’est trop **long ***** car pour nous, c’est dur à vivre.* Two **years **** of psychiatry is too **long ***** because for us, it’s hard to live. (gender diverse; how health professionals can better respond)
	*La file d’**attente *** pour les **opérations *** est trop **longue ***** et la prise de médicaments avec autorisation des parents n’est pas **juste *** pour les mineurs.* The **waiting *** list for **surgery *** is too **long ***** and requiring parental permission to take medication is not **fair *** to minors. (woman; improving information, support, or care provided)
**4. Raise awareness in civil society**	***Informer *******dans les **écoles ***** dès la **primaire **** puis en évoluant avec l’âge.* **Provide information ***** in **schools ***** starting in **primary school **** and progressing with age. (woman; improving information, support, or care provided)
	*Les trans doivent être plus représenté dans les **médias ***** et les dessins animés pour montrer aux enfants afin qu’ils soient plus ouverts.* Trans people need to be more represented in the **media ***** and in cartoons to teach children to be more open. (gender diverse; improving information, support, or care provided)
	***Sensibiliser *******les parents pendant la grossesse.* **Raise** parents’ **awareness ***** during pregnancy. (man; improving information, support, or care provided)
	***Sensibiliser *******à la transphobie dans les **écoles ***** (mise en situation* via *des vidéos comme les spots contre le racisme).* **Raise awareness ***** of transphobia in **schools ***** (using videos such as anti-racism spots). (man; improving information, support, or care provided)
**5. Facilitate the change of civil status**	***Changer *****la **loi ***** pour **rendre ***** le **changement ***** d’état **civil *** libre ***** et **gratuit ***** en **mairie *****.* **Change *** the **law ***** to **make *** civil ***** status **change *** free ***** and **unrestricted ***** at the **town hall *****. (woman; improving information, support, or care provided)
	***Simplifier *******la nouvelle **loi ***** pour la **rendre ***** plus précise pour éviter l’appréciation du juge.* **Simplify ***** the new **law ***** to **make ***** it more precise to avoid the judge’s appreciation. (gender diverse; improving information, support, or care provided)
	*Arrêter de **demander **** des **attestations *****. C’est inutile car on n’est pas malade.* Stop **asking **** for **attestations *****. It’s useless because we’re not sick. (gender diverse; improving information, support, or care provided)
	*Faciliter les **changements ***** administratifs. Les **rendre *** libres ***** et **gratuits *****.* Simplify administrative **changes *****. Make them **free ***** and **unrestricted *****. (woman; improving information, support, or care provided)

Bolded nouns, verbs, and adjectives are statistically associated with the cluster they are presented in (* *p* < 0.05; ** *p* < 0.01; *** *p* < 0.001).

**Table 3 ijerph-19-13257-t003:** Proportion of corpus included and loss of data between analyses.

Clusters	In First Classification (92.9% of Corpus)	In Second Classification (76.5% of Corpus)	Proportion of Corpus	Corpus Lost between Classifications
Cluster 1	27.6%		25.6%	3.2%
Cluster 1.0	4.1%	5.0%	3.8%	
Cluster 1.1	6.5%	7.9%	6.0%	
Cluster 1.2	4.7%	5.7%	4.4%	
Cluster 1.3	4.7%	5.7%	4.4%	
Cluster 1.4	4.1%	5.0%	3.8%	
Total	24.1%	29.3%	22.4%	
Cluster 2	9.4%		8.7%	0.0%
Cluster 2.0	9.4%	11.4%	8.7%	
Cluster 3	20.0%		18.6%	0.5%
Cluster 3.0	10.6%	12.9%	9.9%	
Cluster 3.1	8.8%	10.7%	8.2%	
Total	19.4%	23.6%	18.1%	
Cluster 4	33.5%		31.1%	12.5%
Cluster 4.0	5.3%	6.4%	4.9%	
Cluster 4.1	11.2%	13.6%	10.4%	
Cluster 4.2	3.5%	4.3%	3.3%	
Total	20.0%	24.3%	18.6%	
Cluster 5	9.4%		8.7%	0.0%
Cluster 5.0	9.4%	11.4%	8.7%	

## Data Availability

Data were accessed by a convention with the World Health Organization Center for research and training in mental health (EPSM Lille-Métropole) which does not allow public release of materials. Original study data can be accessed by requesting the owner.

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
