# Peer review of "Impact of Depsychopathologization of Transgender and Gender Diverse Individuals in ICD-11 on Care Delivery: Looking at Trans Expertise through a Trans Lens"

_ijerph, 2022, doi:10.3390/ijerph192013257_

Round 1

Reviewer 1 Report

It is simply an excellent and highly academic contribution.

First of all, thanks for this transgender expertise through a trans lens; that is too sadly too seldom yet, that we have to the opportunity to have a better scrutiny on informed consent-based health care, especially when crossed with gender studies.

I wish we could have more input about intersex individuals in future researches, hopefully.

Very good analysis of political dynamics concerning gender studies epistemology, I wish to read more about that in the future.

Author Response

We thank you for your enthusiastic feedback and interest in our work.
We have improved the overall English level following your recommendations.
Research involving intersex people, and particularly the intersections with transgender and human rights, is indeed a necessary area of research, especially participatory research. Unfortunately, analysis using routine data remains difficult due to administrative and social organization barriers and research is underfunded.
However, we hope to be able to explore this aspect further in the future, in conjunction with concerned organizations.

Reviewer 2 Report

I have reviewed the manuscript entitled “Impact of depsychopathologization of transgender and gender 2 diverse individuals in ICD-11 on care delivery: looking at trans 3 expertise through a trans lens.” It can be said that this paper draws attention to a very important issue. However, I recommend that the authors carefully examine the points I have listed below.

1.       I suggest editing the Abstract. For example, the number of participants must be specified.

2.       Mentioning the purpose of the study in the first paragraph of the Introduction gives the impression that this part was written inexperienced. Please shift the part about the purpose to the last paragraph of the Introduction.

3.       In the Introduction, it would be appropriate to mention both the positive and negative aspects of depsychopathologization.

4.       The authors mentioned that the lack of representation of TGD individuals in studies creates a lack of quality data. Is there convincing evidence for this? According to the authors, why are studies conducted by individuals without TGD unable to provide quality data?

5.       What are the hypotheses of the study?

6.       The purpose of the article is not fully understood. "French study" --> which study? “shared knowledge” --> what /which knowledge?

7.       It can be said that it would be important to briefly mention the study carried out collaboratively with the WHO.

8.       “They were interviewed using two questionnaires: the main questionnaire translated and adapted from the one used in Mexico [11], and an additional one specific to the French study” -->  It is not fully understood what these questionnaires are and what they evaluate. Please provide more detailed information (including names).

9.       It is not clear how contact was established with the participants. Please explain.

10.   In the Discussion section, it is seen that the findings are not sufficiently discussed, rather the findings are written in order. In this section, please discuss the results of the study in the context of the relevant literature.

Author Response

We thank you for your feedback on our work. Here is a point-by-point response to your comments:

- We improved the overall English level, as suggested;

- We edited the abstract to include the number of participants in the study;

- We changed the structure of the introduction and moved the purpose of the study to the end;

- We took your comment about the positive and negative aspects of depsychopathologization into consideration and understand the value of situating it in the current research landscape. We have chosen to focus on the epistemic impact rather than on its positive and negative aspects. This perspective seemed more relevant since, as suggested in our references, this change in the way transgender people are viewed in health is rooted more in a shift in perspective (with paradigmatic and ideological ramifications) than in a natural evolution of previous concepts. We expanded on this perspective - the main contribution of our work - in the discussion;

- We illustrated two major stakes regarding the effective participation of transgender people in research and developed the benefit of participative research in the discussion. Between the references, the synthetic approach in the introduction, a dedicated paragraph in the methodology, and a better discussion, we think this aspect is sufficiently covered given the state-of-the-art in the field. Again, we thank you for highlighting this point;

- We formalized the main hypothesis in a more obvious way and integrated it into the overall epistemic discussion;

- We clarified some terms like 'French study' and 'shared knowledge'. Thank you for bringing these possible confusions to our attention, we believe that they are the result of the close proximity we have with our research topic and these modifications will help to make our paper more accessible;

- We summarized the main results of the studies conducted with WHO and added references to a related paper;

- We further clarified the role of each questionnaire and how they were used;

- We provided more detail on participant recruitment methods;

- As noted above, we enhanced our discussion with an epistemic consideration, one of the two major strengths of this paper - the other being documenting feedback from non-affiliated transgender people in a free and informed consent context that the WPATH cited as an example model for reorganization in the SOC8.

We hope these changes address your feedback in a satisfactory manner. We believe they contribute to a better quality of the paper by further anchoring it in the current scientific landscape. We thank you for your insightful feedback.

Round 2

Reviewer 2 Report

The authors duly implemented the revisions. This paper is in a much better condition.

Minor suggestion; 

Do the questionnaires applied have names? If there is, I suggest that these names be written in the manuscript.